# Combined Ammonia and Electron Processing of a Carbon-Rich Ruthenium Nanomaterial Fabricated by Electron-Induced Deposition

**DOI:** 10.3390/mi11080769

**Published:** 2020-08-12

**Authors:** Markus Rohdenburg, Johannes E. Fröch, Petra Martinović, Charlene J. Lobo, Petra Swiderek

**Affiliations:** 1Institute for Applied and Physical Chemistry (IAPC), Fachbereich 2 (Chemie/Biologie), University of Bremen, Leobener Str. 5 (NW2), 28359 Bremen, Germany; p.martinovic@uni-bremen.de; 2School of Mathematical and Physical Sciences, University of Technology Sydney, Ultimo, NSW 2007, Australia; Johannes.E.Frch@student.uts.edu.au (J.E.F.); Charlene.Lobo@uts.edu.au (C.J.L.)

**Keywords:** focused electron beam-induced deposition, deposit post-processing, ammonia, Ru deposition, carbon nitride

## Abstract

Ammonia (NH_3_)-assisted purification of deposits fabricated by focused electron beam-induced deposition (FEBID) has recently been proven successful for the removal of halide contaminations. Herein, we demonstrate the impact of combined NH_3_ and electron processing on FEBID deposits containing hydrocarbon contaminations that stem from anionic cyclopentadienyl-type ligands. For this purpose, we performed FEBID using bis(ethylcyclopentadienyl)ruthenium(II) as the precursor and subjected the resulting deposits to NH_3_ and electron processing, both in an environmental scanning electron microscope (ESEM) and in a surface science study under ultrahigh vacuum (UHV) conditions. The results provide evidence that nitrogen from NH_3_ is incorporated into the carbon content of the deposits which results in a covalent nitride material. This approach opens a perspective to combine the promising properties of carbon nitrides with respect to photocatalysis or nanosensing with the unique 3D nanoprinting capabilities of FEBID, enabling access to a novel class of tailored nanodevices.

## 1. Introduction

Nanoscale materials fabricated by focused electron beam-induced deposition (FEBID) drive the development of novel devices for a variety of applications [1,2,3]. In FEBID, volatile precursor molecules that contain an element of the desired material are adsorbed on a surface and fragmented under a tightly focused electron beam. In this process, a solid deposit is formed, while volatile fragments desorb and are pumped away. However, the electron-induced fragmentation of organometallic precursors used to produce metallic materials is often incomplete and less volatile ligands remain behind, leading to strongly contaminated deposits [1,4,5]. Therefore, purification processes have been devised, in which unwanted elements are removed by thermal treatment, continued electron irradiation, reactions with process gases, or a combination thereof [4].

As a particularly successful and mild post-deposition purification process, electron irradiation in the presence of H_2_O vapor was recently brought forward. This process was applied to deposits produced from the precursors trimethyl(methylcyclopentadienyl)-platinum(IV) (MeCpPtMe_3_) [6], tetrakis(trifluorophosphine)platinum(0) (Pt(PF_3_)_4_) [7], dimethyl(acetylacetonate) gold(III) (Me_2_Au(acac)) [8], and, more recently, from bis(ethylcyclopentadienyl)ruthenium(II) ((EtCp)_2_Ru) [9]. In all cases, the carbon residue was efficiently removed and it was demonstrated that even the shape of complex 3D nanostructures can be preserved during such treatment [8]. In a related process, dimethyl(trifluoroacetylacetonate)gold(III) and H_2_O vapor were injected simultaneously during the FEBID process, again yielding a high-purity Au deposit [10]. In addition, H_2_O was used under electron irradiation to etch away hexagonal boron nitride (hBN) for fabrication of silver nanowire–nitride heterostructures on surfaces [11].

While FEBID generally strives to produce highly pure metals, impure deposits may also have interesting properties. Such deposits are often nanogranular materials consisting of metal nanoparticles embedded in an amorphous carbonaceous matrix [6,9,12]. For instance, Pt in a carbonaceous matrix deposited from MeCpPtMe_3_ can serve as humidity [13] or strain sensing material [14]. This function relies on the electric conductivity of the material which is governed by tunneling of the charge between the metal grains [1]. The height of the barrier that determines the efficiency of this transport process depends on the distance between the metallic grains which is altered by mechanical strain [1], but can also be influenced by contact of H_2_O with the material, which is exploited to measure humidity [13].

It is obvious that the efficiency of the latter type of sensing by such nanogranular materials depends on the uptake of the analyte by the carbonaceous matrix. This property can be tuned by providing stronger binding sites which, for instance, can be achieved by incorporating nitrogen in the carbon network. Following this strategy, amorphous carbon nitride has been suggested as a low-cost and environmentally friendly material for gas sensing [15]. While research on carbon nitride materials has predominantly focused on crystalline forms and their applications as emerging materials for semiconducting devices, photocatalysis, energy storage devices, catalyst supports or catalysts [16,17,18], amorphous carbon nitrides are particularly easy to produce [15].

The useful properties of carbon nitrides may open new perspectives when transferred to nanoscale devices in a direct write approach. In analogy to the diverse synthetic strategies for bulk carbon nitride materials by polymerization of N-containing heterocycles [16], this was previously attempted by electron-induced crosslinking of 1,2-diaminopropane (1,2-DAP) [19]. However, amines tend to be difficult to handle in vacuum systems. Here, we demonstrate that electron irradiation, in the presence of ammonia (NH_3_), performed on carbon-rich deposits produced from the organometallic precursor (EtCp)_2_Ru, converts the carbonaceous matrix to a material with the typical composition of carbon nitride (C_3_N_4_). This study builds on our previous work regarding the electron-induced chemistry of NH_3_. With respect to FEBID applications, we demonstrated that chlorine-contaminated ruthenium deposits produced from η^3^-allyl ruthenium tricarbonyl chloride (η^3^-C_3_H_5_)Ru(CO)_3_Cl can be purified by electron exposure in the presence of NH_3_ [20]. Additionally, NH_3_ enhances the rate of metal reduction when applied as a precursor ligand [21,22]. However, NH_3_ also adds to unsaturated hydrocarbon structures in an electron-induced reaction [23,24,25]. This latter aspect, in particular, is exploited here and opens a novel pathway to the nanoscale fabrication of carbon nitride devices by FEBID.

For this contribution, we performed processing of FEBID deposits produced from (EtCp)_2_Ru via high-energy electron irradiation in the presence of NH_3_ in an electron microscope and analyzed the resulting material in terms of chemical composition and deposit morphology, employing energy-dispersive X-ray spectroscopy (EDX) and atomic force microscopy (AFM). In addition, we performed a surface science study on model deposits fabricated under cryogenic conditions and investigated the impact of simultaneous electron beam and NH_3_ processing using a combination of reflection–absorption infrared spectroscopy (RAIRS), Auger electron spectroscopy (AES), and methods relying on mass spectrometry (MS), namely thermal desorption spectrometry (TDS) and experiments on electron-stimulated desorption (ESD) monitoring volatile irradiation products.

## 2. Materials and Methods

Electron beam irradiation was performed in an environmental scanning electron microscope (ESEM): the FEI Nova NanoSEM (FEI Company, Eindhoven, The Netherlands) with a background pressure of 3 × 10^−6^ mbar. Unless otherwise stated, FEBID was performed using a 5 keV beam at a beam current of 500 pA on Si substrates with a native SiO_2_ layer (SiO_2_/Si). A focused Gaussian electron beam was rastered over the region of interest and the irradiation process was recorded in real time. Compositional analysis was conducted by EDX (Oxford Instruments INCA x-sight, Oxford Instruments, Abingdon, UK) spectroscopy after processing by exposing an area of 500 × 500 nm^2^ to a 5 keV electron beam at a beam current of 300 pA while collecting ejected X-rays for 1 min. Characterization of deposit morphology was conducted using tapping mode AFM (Digital Instruments Dimension 3100, Digital Instruments, Bresso, Italy) at intervals during the FEBID process.

The surface science experiments were performed in an ultrahigh vacuum (UHV) chamber with a base pressure of about 10^−10^ mbar [26]. It contains a polycrystalline Ta sheet used as substrate held at 110 K by liquid N_2_ cooling. The sample temperature is controlled by resistive heating of two thin Ta ribbons spot-welded to the thicker Ta sheet and is measured using a type E thermocouple press-fitted to the Ta substrate. The setup is equipped with a quadrupole mass spectrometer (QMS) residual gas analyzer (300 amu, Stanford Research Systems, Sunnyvale, CA, USA) with electron impact ionization at 70 eV, a commercial flood gun (SPECS FG 15/40, SPECS Surface Nano Analysis GmbH, Berlin, Germany) for electron irradiation in the range of *E*_0_ = 1–500 eV, an Auger electron spectrometer (STAIB DESA 100, STAIB INSTRUMENTS GmbH, Langenbach, Germany), and a sputter gun operated with Ar^+^ ions at a kinetic energy of 3 keV. All AE spectra were recorded using an electron energy of 5 keV in derivative acquisition mode. The infrared beam of a commercial RAIR spectrometer (IFS 66v/S, Bruker Optics GmbH, Ettlingen, Germany) is guided through a KBr window into the UHV chamber, allowing for vibrational spectroscopy of adsorbates on the Ta sheet under grazing incidence conditions. The RAIR spectrometer is equipped with a liquid nitrogen-cooled MCT detector with a sensitivity limit down to 750 cm^−1^. RAIR spectra were collected from 4000 to 750 cm^−1^ with a resolution of 4 cm^−1^ by averaging 400 single scans. The optics system was continuously purged with N_2_ to eliminate contributions of residual vapors to the RAIR spectra.

Model deposits were fabricated from (EtCp)_2_Ru using a slightly modified protocol, as described in reference [9]. Briefly, precursor vapor was leaked into the UHV chamber via a gas handling manifold, where the pressure was measured with a Baratron pressure transducer with mTorr reading. An amount of precursor vapor corresponding to a pressure drop of 2 mTorr in the manifold was condensed onto the cooled Ta sheet, resulting in a coverage of 7–10 monolayers (ML) [9]. The film was then irradiated at *E*_0_ = 31 eV with a total dose in the range 40–60 mC/cm^2^. For a dose of 40 mC/cm^2^, ESD of desorbing products had ceased according to our recent study [9]. Material retained under cryogenic conditions was subsequently desorbed by a TDS run and a bakeout (450 K for 30 s). Products that are observed during these ESD and TDS procedures are described in detail in [9]. After the Ta sheet with the remaining material had cooled back to 110 K, a film of NH_3_ corresponding to a pressure drop of 4 mTorr in the gas handling manifold was condensed on top of the model deposit. The NH_3_ film thickness was estimated to be at least in the range of a monolayer in accordance with [20]. The resulting layer was then again irradiated at *E*_0_ = 31 eV with a total dose of 40 mC/cm^2^ while monitoring RAIRS and ESD. After irradiation was terminated, a subsequent TDS run followed by a bakeout (450 K, 30 s) removed from the surface the remaining material that was retained at 110 K. This cycle of NH_3_ dosing, irradiation at *E*_0_ = 31 eV, and final TDS with bakeout was repeated several times. After some of these NH_3_ processing cycles, AES was performed to monitor the elemental composition of the remaining material on the Ta sheet.

## 3. Results

### 3.1. Deposit Formation, Combined Electron and NH_3_ Processing, and Post-Processing Analysis

FEBID was used to fabricate square-shaped deposits from the precursor (EtCp)_2_Ru on a SiO_2_/Si substrate. The deposits had a height of 270 nm as observed by both SEM and AFM imaging (see Figure 1, left side, and Appendix A). The chemical composition of the deposits was determined by EDX as C 87.1%, Ru 9.7%, and O 3.2%. The latter signal can be ascribed to O from the SiO_2_ native layer of the substrate. However, the C:Ru ratio of ~9:1 agrees with former EDX studies of the composition of FEBID deposits fabricated from (EtCp)_2_Ru [27].

Combined electron irradiation and NH_3_ processing was carried out by scanning the 5 keV electron beam in a 1.2 × 1.2 µm^2^ region around the deposits in a background chamber pressure of 0.11 mbar NH_3_. According to SEM and AFM, the deposit height significantly increased from initially 270 to 480 nm (+~80%) after 30 min of processing. This result is in contrast to analogous postprocessing of (EtCp)_2_Ru deposits with H_2_O instead of NH_3_. In this previous study, the deposit height decreased strongly after short purification (13 s at 5 keV and 5.6 nA for a 500 × 500 nm^2^ deposit) in the presence of ~0.5 mbar H_2_O and 5 keV electron irradiation [9]. Rather than removing material, processing in NH_3_ leads to incorporation of the process gas into the deposit and consequent swelling, as is obvious from the occurrence of large bubble-type structures in the halo regions of the deposits (see SEM and AFM images in Figure 1, right side). Electron-induced fragmentation of NH_3_ that has diffused into the deposit yields N_2_ (see also ESD experiments in Section 3.2.2) and releases atomic hydrogen (AH) that can, among other reactions, recombine to form H_2_ [22,25,28]. These gases can be held responsible for the observed swelling and bubble formation.

To evaluate changes in the chemical composition of the deposited material under combined electron beam and NH_3_ processing, we performed EDX after several stages of purification. This was done by exposing a set of FEBID deposits with comparable size and thickness (Appendix A) to varying NH_3_ processing durations in the presence of the 5 keV electron beam. The results are depicted as circles in Figure 2a. NH_3_ post-processing for less than 10 min had no effect on either material composition or deposit height as obvious from AFM images (see Appendix A). After 10 min of post-processing, however, a decrease in the relative C (−17% compared to the as-deposited composition) and Ru (−3.2%) signals was observed, accompanied by an increase in the N signal (+18.5%) and a minor increase in O (+1.1%). As the height gain after 10 min is negligible (see Appendix A), the chemical changes only relate to the deposit volume and not to a variation in the contribution of the underlying substrate to the EDX spectra. The rapid decrease in the C signal and incorporation of N continued until 30 min of processing and subsequently, declined. Finally, the longest processing time of 60 min resulted in a deposit with a maximum height of ~600 nm (see Appendix A) consisting of 37.5% C (−49.6%), 49.0% N, 8.3% Ru, and 5.2% O (circles in Figure 2a). The Ru and O content was, thus, nearly unaffected by the combined electron beam and NH_3_ processing. However, the decreased C:Ru ratio indicates that C was lost from the deposit and replaced by N. Note that the C:N ratio of the obtained material is that of a typical carbon nitride, i.e., C_3_N_4_ [16,17,18]. We note that the density of amorphous C_3_N_4_ (1.3–1.4 g/cm^3^ [29]) as compared to that of a RuC_9_ deposit (3.2 g/cm^3^ [9]) can account for a volume increase of roughly 230–250%. A height increase of ~220% (see Appendix A) compares well with this estimate. The lateral growth after processing as clearly seen in Figure 1 (top), however, shows that additional swelling due to evolution of gas must be involved in the overall volume increase. We additionally note that electron-induced reactions of NH_3_ with the Si substrate are unlikely to significantly contribute to the increase in the N EDX signal. In a control experiment in absence of a (EtCp)_2_Ru deposit, only a small N signal was detected after 30 min of processing (see Appendix A).

EDX mapping (Figure 2b) of a deposit that was post-processed for 30 min reveals that the distribution of C, Ru, and N reflects the egg-shaped thick central area. SEM imaging shows that processing has generated a hole in the middle of the deposit that is also visible in the EDX map. In contrast, O is seen mainly in the outer regions where the Si substrate contributes to EDX intensities.

The effect of the combined electron beam and NH_3_ processing was also studied on (EtCp)_2_Ru deposits that had been pre-purified by electron irradiation in the presence of H_2_O [9]. In this case, the carbon content was significantly reduced before NH_3_ processing. After H_2_O-assisted purification (5 keV, 1 nA, 0.13 mbar H_2_O, 30 min), the deposits typically contained 47.9% Ru, 43.8% C, and 8.3% O (triangles in Figure 2a). NH_3_ post-processing of such pre-purified deposits yielded comparable results to unpurified deposits: 60 min of processing resulted in a composition of 12.2% C (−31.6%), 20.4% N, 51.0% Ru, and 16.3% O. The carbon content was, thus, again reduced at the expense of N intake into the deposit. Note that the carbon content after an equal processing time of 60 min was lower when the deposits were pre-purified by electrons in the presence of H_2_O (12.2% C) as compared to those that were not pre-purified (37.5% C). This also results in a different C:N ratio of the processed deposits (with pre-purification: 0.60, without pre-purification: 0.77). Carbon content and C:N ratio in the deposit can, thus, be tuned by adjusting the processing conditions. Similar to deposits that were not pre-purified in the presence of H_2_O, a height increase from initially ~100 nm (after H_2_O-assisted purification) to a maximum of ~600 nm was observed by AFM (see Appendix A). In terms of EDX elemental mapping (Figure 2c), the highest Ru concentration was found in the central region where the purification yielded a densified Ru deposit with reduced C content. C is also predominantly observed in this central region. The N signal, however, is spread across the whole processing area, pointing to incorporation of N both in the central deposit area and in the thinner halo region. In line with the mild character of the H_2_O-assisted purification [9], O was not very pronounced in the central region of the deposit but was concentrated in the halo and the surrounding area where the Si substrate contributes to the EDX intensities.

### 3.2. Surface Science Study

#### 3.2.1. Formation of Model Deposits

A surface science study was performed to elucidate the chemical form of N incorporated into the (EtCp)_2_Ru deposits during the combined electron beam and NH_3_ processing. As a model system for FEBID deposits, thin layers (7–10 ML corresponding to an average thickness of 7–10 nm [9]) of (EtCp)_2_Ru were condensed on a cryogenic surface (110 K), extensively irradiated (*E*_0_ = 31 eV, 40–60 mC/cm^2^), and subsequently, annealed (450 K) [9]. The products that desorb during both electron exposure and annealing have been described in detail elsewhere [9]. Briefly, methane (CH_4_), ethene (C_2_H_4_), and ethane (C_2_H_6_) are lost during ESD, along with a small fraction of Cp-containing species (see Appendix A for a thicker 13–20 ML precursor film). This accounts for the high carbon content of as-deposited material from (EtCp)_2_Ru. It is clear that most of the observed products relate to fragmentation of the Et side chains of the EtCp ligands. All ESD signals leveled off after an exposure of 40 mC/cm^2^ [9]. An exposure in the range 40–60 mC/cm^2^ in combination with an even thinner precursor film is, thus, sufficient to convert the molecular precursor film into a model deposit. The persistent Cp moieties of the precursor become embedded into the deposit upon electron exposure. This is also obvious from the reduced intensity of the precursor desorption signal at ~210 K in post-exposure TDS as compared to a non-irradiated film (see Appendix A) [8]. In addition, some further C2 fragments desorb between 100 and 150 K. Note that this desorption temperature is higher than observed previously for condensed mixtures of C_2_H_4_ (about 65 K), C_2_H_6_ (about 70 K) and NH_3_ (around 100 K) [25]. The delayed desorption of the C2 products in post-exposure TDS of (EtCp)_2_Ru must, thus, be ascribed to trapping of the small hydrocarbons in the model deposit.

In the present study, the formation of the deposit was also monitored by RAIRS, which shows the CH stretching bands of the saturated Et side chains between 2800 and 3000 cm^−1^ as the most intense signals (Figure 3). In line with the layer thickness of 7–10 nm [9] and the fact that the typical mean free path of 31 eV electrons in condensed material is below 1 nm [30], some precursor in deeper layers is left intact after an exposure of 40 mC/cm^2^, as obvious from the remaining intensity of the CH stretching bands. After TDS, the remaining intensity was strongly reduced, and the band shape altered, pointing to removal of the intact precursor molecules that were not decomposed by electron irradiation.

#### 3.2.2. First Cycle of Electron Exposure in Presence of NH_3_

When NH_3_ is adsorbed on the deposit and electron exposure at *E*_0_ = 31 eV is resumed, MS acquired during ESD (Figure 4) shows dominant signals of NH_3_ (*m*/*z* 17) and N_2_ (*m*/*z* 28), the latter pointing to the electron-induced release of hydrogen [20,22]. Note that the intensity at *m*/*z* 16 (intensity ratio *m*/*z* 16:*m*/*z* 17 = ~0.8) is not enhanced as compared to the reference MS of NH_3_ [32]. Therefore, and in contrast to our previous study on ESD from cisplatin, there is no evidence for desorption of NH_2_^•^ radicals during processing. Additional low-intensity signals at *m*/*z* 26–30 are present, indicating that more of a C2 product is formed. Here, the presence of *m*/*z* 30 points to C_2_H_6_ [32] in line with the reducing action of NH_3_. Intensity at *m*/*z* 30 could also be related to diazene (N_2_H_2_) desorption, but it would be surprising to detect such an unstable species but not the more stable NH_3_ irradiation product hydrazine (N_2_H_4_) at *m*/*z* 32 [33].

In Figure 5, RAIR spectra are shown for a NH_3_ film that has been condensed on top of an (EtCp)_2_Ru model deposit at *T* = 110 K in the pristine state (top), after several stages of electron exposure at *E*_0_ = 31 eV (middle), and after a final thermal annealing step to *T* = 450 K in the course of a TDS run (bottom). NH_3_ vibrations are marked with dashed black lines: At 3382 cm^−1^, the asymmetric NH_3_ stretch is observed, along with an asymmetric (1622 cm^−1^) and two symmetric deformations (1121 and 1072 cm^−1^). The band locations are in good agreement with assignments published for NH_3_ condensed on a highly ordered pyrolytic graphite (HOPG) surface [34]. The symmetric NH_3_ stretch at 3299 cm^−1^ [34] is overlaid with O–H stretches stemming from water ice frozen onto the entry window of the cooled infrared detector (all H_2_O-related vibrations [35] are marked with dashed blue lines in Figure 5).

Upon electron exposure at *E*_0_ = 31 eV, all NH_3_-related bands decreased in intensity in line with efficient ESD of NH_3_, as observed previously at even lower *E*_0_ than applied here [25]. Additionally, negative signals in the range of aliphatic CH stretches (2800–3000 cm^−1^), and CC stretches/CH_3_ bending vibrations (1200–1500 cm^−1^) evolved as the electron dose was increased. Since the state after thermal annealing of the model deposit and just before NH_3_ dosing is reflected in the background that was employed for all spectra, these negative signals indicate loss of CH and CC stretches while NH_3_ is consumed. Furthermore, post-NH_3_-processing TDS (see Appendix A) shows again some desorption of a product with a *m*/*z* 28 fragment between 100 and 150 K. This is possibly also a C2 product, but some contribution of N_2_ trapped in the model deposit cannot be ruled out. In addition, we observed a weak vibrational feature at ~1044 cm^−1^ after processing with 10 mC/cm^2^ for all higher electron doses and after TDS (see inset in Figure 5). The position of this feature coincides with the position of the CN stretching in methylamine [36,37]. The CN stretching vibration of saturated amines is typically located in the range 1020–1250 cm^−1^ [38]. Combined NH_3_ and electron processing of deposits fabricated from (EtCp)_2_Ru, thus, not only leads to further removal of hydrocarbons but also to the formation of C–N bonds.

#### 3.2.3. Effect of Repeated Processing Cycles of Electron Exposure in Presence of NH_3_

Beyond the first processing cycle, the chemical composition of the (EtCp)_2_Ru model deposits was monitored by AES. The spectra were acquired after different numbers of processing cycles consisting of NH_3_ dosing and electron irradiation at *E*_0_ = 31 eV (40 mC/cm^2^) at 110 K and a subsequent TDS run (Figure 6a).

Unfortunately, the most intense Ru signal in AES (MNN at 277 eV [39]) and the C KLL signal (275 eV [39]) are located too close to each other to be resolved by our instrument. Therefore, a broad signal accounting for both species is the only feature observed for the as-deposited model deposit. With ongoing processing, this signal did not significantly decrease in intensity but a new feature at 386 eV appeared, indicative of N [39]. The N KLL signal in AES is a sensitive probe for the chemical state of N bound to another element. An earlier study has compared the N KVV signals in AES measured in differential mode from diverse N-containing elemental layers [40]. While the exact position of the N KVV bands shifted slightly with increasing N content, as exemplified for Ru-N samples, it was shown that the fine structure of the bands is characteristic of the element in which N was incorporated [40]. In particular, both Ta–N and Ru–N revealed narrow distances between the minima and maxima, while covalent nitrides such as C–N exhibited wide peak structures [40]. For the present AES data (Figure 6b), the distances between the main maxima are on average 13 eV in the case of Ta reacted with NH_3_, in excellent agreement with the spectrum reported in reference [40], where a similar line shape was also observed for Ru–N. In contrast, the AES data obtained after processing of the carbonaceous Ru model deposits (Figure 6a,b) revealed a spacing near 20 eV between the two main maxima, this time in excellent agreement with the previous data for C–N samples [40]. This clearly shows that N from the adsorbed NH_3_ is incorporated in the C residue during electron exposure and does not react with Ru.

## 4. Discussion

We now propose reaction mechanisms that can rationalize the chemical composition of FEBID deposits produced from (EtCp)_2_Ru, as observed after electron beam processing in the presence of NH_3_.

Based on the observation that N_2_ is released in ESD experiments (Figure 4) and in line with the fact that NH_3_ acts as a reducing agent under electron irradiation [20,22], NH_3_ must be decomposed under electron impact yielding AH. In addition, ionization of NH_3_ can lead to a proton transfer reaction as predicted by theory [41] and observed in experiments [42] for the ionized (NH_3_)_2_^•+^ dimer (Equation (1)).
(NH_3_)_2_^•+^ ⟶ NH_4_^+^ + NH_2_^•^(1)

Electron exposure of (EtCp)_2_Ru mainly leads to cleavage of the Cp-Et bonds, resulting in loss of C2 hydrocarbons from the precursor (see Section 3.2.1 and reference [9]). As a simplified model that reflects the composition of deposits fabricated from (EtCp)_2_Ru, we therefore employ ruthenocene (Cp_2_Ru) in all following discussions. When a proton from NH_3_^•+^ is transferred to a Cp_2_Ru molecule instead of a neighboring NH_3_, the reaction would lead to cleavage of the metal–ligand bond, since cyclopentadienyl ligand-carrying precursors tend to lose their Cp ligands when they become protonated [43,44]. Equation (2) represents this reaction in the case of Cp_2_Ru.
NH_3_^•+^ + Cp_2_Ru ⟶ CpH + CpRu^+^ + NH_2_^•^(2)

In the following, we explain how the products of Equation (2), namely cyclopentadiene (CpH, **1**), cyclopentadienyl ruthenium(II) cations (CpRu^+^), and amino radicals (NH_2_^•^) possibly contribute to the formation of carbon nitrides as observed after processing of the (EtCp)_2_Ru deposits.

Combined ESD and TDS results of the surface study (see Section 3.2.1 and reference [9]) as well as EDX analysis in electron microscopes (see Section 3.1 and reference [27]) suggest that most of the Cp moieties of the EtCp ligands become embedded in the deposits upon electron exposure. The majority of **1** as the most volatile, small hydrocarbon product of Equation (2) is, thus, retained in the deposit. The cycloaddition of dienes via single electron oxidation and subsequent formation of the Diels–Alder dimer has been described in the literature [45]. Due to the presence of C=C double bonds even after dimerization, the resulting Diels–Alder adduct is prone to further crosslinking upon prolonged electron exposure, producing a larger, non-volatile hydrocarbon residue.

Aside from crosslinking, the C=C double bonds can be subject to an electron-induced hydroamination reaction in the presence of NH_3_. This reaction has been previously observed in mixtures of various alkenes and NH_3_ [23,24,25]. Electron-induced hydroamination is initiated by electron impact ionization (EI) of either the alkene or NH_3_, leading to attractive forces between the resulting radical cation and its electron-rich reaction partner [23]. This produces a radical cation adduct in the deposit that can subsequently be quenched by a thermalized electron. Scheme 1 shows this electron-induced hydroamination of **1** as our simplified model for chemical motifs in the hydrocarbon residue. The reaction product shown in Scheme 1 is cyclopent-2-en-1-amine (**2**). Note that electron-induced hydroamination reactions are hardly regioselective [25] so that cyclopent-3-en-1-amine (**3**) would be another conceivable reaction product.

The NH_2_^•^ radicals produced according to Equations (1) and (2) may also add to C=C double bonds present in the hydrocarbon residue. This results in direct amination of these moieties. The resulting amine radical can be saturated by addition of AH. Note that radical addition can initiate a chain reaction involving radical site propagation and, thus, extends the degree of crosslinking in the deposit. Furthermore, in alkenes with allylic H atoms, substitution of one of these H atoms by a radical under release of AH is another conceivable reaction. In the case of **1**, this reaction would result in formation of cyclopenta-2,4-dien-1-amine (**4**). Scheme 2 visualizes the reaction pathways involving NH_2_^•^ radicals.

The RAIRS investigation of the first processing cycle (Figure 5) was carried out in situ, i.e., by keeping the sample continuously under UHV conditions. In our previous study on the electron-induced decomposition of cisplatin, we identified NH_2_^•^ radicals as intermediates in an in situ X-ray photoelectron spectroscopy (XPS) study, but not in a complementary RAIRS study that was carried out ex situ [22]. The absence of vibrational features of the NH_2_^•^ moieties in ex situ RAIRS was explained by consumption of the reactive radicals during handling at ambient conditions [22]. The absence of NH_2_^•^ vibrational features like δ_w_(NH_2_) at 1392 cm^−1^ or δ_s_(HNH) at 1555 cm^−1^ [46] in in situ RAIRS in the present study, thus, points to a rapid consumption by the described reactions with the hydrocarbon residue. The weak C–N stretching feature present after deposit processing (Figure 5) together with the absence of NH_2_^•^ desorption in ESD (Figure 4) and the AES data (Figure 6b) indicate that the fate of the NH_2_^•^ radicals must indeed be conversion into carbon-bound moieties.

After EI, cyclic C5 amines, as can be formed via the reactions proposed above, typically fragment via ring-opening α cleavage followed by H rearrangement [47]. The EI-MS of **2**, **3**, or **4**, however, has not been reported to the best of our knowledge. We, therefore, briefly discuss the major dissociation channel following EI of cyclopentanamine (C_5_H_11_N, M^•+^ at *m*/*z* 85) as a simple example for fragmentation of such cyclic C5 amines. The EI-MS of cyclopentanamine exhibits the most intense signal at *m*/*z* 56 [32]. This fragment results from the loss of a C2 radical from M^•+^ [32]. Scheme 3 shows how this kind of fragmentation following EI would proceed in **2**. The loss of C2 radicals yields new CC double or triple bonds that may either crosslink or be subject to hydroamination under prolonged electron exposure in the presence of NH_3_.

The C2 radicals released during fragmentation, i.e., vinyl (C_2_H_3_^•^) and ethyl (C_2_H_5_^•^) radicals in the case of **2**, may react with AH produced through NH_3_ decomposition to yield C_2_H_4_ and C_2_H_6_, contributing to ESD intensities in the range *m*/*z* 26–30 (Figure 4). Recombination of C_2_H_3_^•^ with NH_2_^•^ radicals would yield ethanimine (**7**) or its tautomer vinylamine (**8**), respectively (Scheme 4). It is known that **7** trimerizes at temperatures above 230 K to yield the acetaldehyde ammonia trimer (**9**) [48]. This reaction could take place during room temperature processing in the electron microscope as well as during annealing of the model deposits after processing and would yield again larger, nitrogen-containing hydrocarbon species that are very likely retained in the processed deposit. Recombination of C_2_H_5_^•^ with NH_2_^•^ would accordingly yield ethylamine (C_2_H_5_NH_2_). In previous studies, it was argued that radical densities in molecular films during electron exposure with the same electron gun as used herein are low and radical recombination reactions are, thus, unlikely to occur [49]. The observation of a hydrocarbon residue with high nitrogen content along with the presence of C2 hydrocarbons in ESD (Figure 4) during processing does not allow us to fully rule out such radical recombination reactions. The initiation of chain reactions by C_2_H_3_^•^ and C_2_H_5_^•^ is, however, a more likely scenario.

The iminium cations **5** and **6**, which are the fragmentation products in Scheme 3, could protonate further Cp moieties in the deposit. Note again that **2** is just a simplified model compound representing likely structural elements in the crosslinked carbon residue of the deposit after electron-induced hydroamination. Therefore, various similar small C_x_N_y_ fragments may be the outcome of the proposed reaction sequence. Overall, electron-induced crosslinking and further NH_3_ addition to these unsaturated species can explain the increasing N content of the non-volatile residue remaining after processing of (EtCp)_2_Ru deposits.

Aside from fragmentation after EI, amines can also undergo crosslinking initiated by dissociative electron attachment (DEA). For instance, it was previously proposed that electron attachment to 1,2-DAP leads to an electronically excited anion, which relaxes via dehydrogenation leaving behind C- and N-centered radicals that can crosslink the molecular film [19]. Such reactions can also explain why volatile amines like ethylamine are not observed in ESD during processing because similar DEA resonances associated with H^−^ loss were found for aliphatic amines like *n*-propylamine [50,51]. Since primary electrons with energies above the ionization threshold such as those used in the present study generate low-energy secondary electrons (LESEs) upon interaction with solid matter, initiation of DEA to amine species in the deposit may likely contribute to the observed crosslinking after processing.

Finally, CpRu^+^ as a further reaction product in Equation (2) is an example of a cationic species that exposes a metal center where additional reactions can occur. For instance, it has been reported that C–N bond formation between ethene and an NH_2_ moiety is facilitated in a cationic Ni(II) amide complex [52]. Such amide complexes may form by addition of NH_2_^−^ anions to an accessible Ru(II) center in CpRu^+^. Note that NH_2_^−^ anions are a major product of DEA to cyclic amines like reported for cyclopropylamine [53] as well as for NH_3_ [54,55,56]. Reactions occurring at accessible Ru(II) centers in the deposit and triggered by LESEs, thus, likely enable formation of new C–N bonds and also rearrangement of amines already present in the residue according to the reactions proposed in Scheme 1 and Scheme 2. Such reactions are of special relevance, since it was reported that NH_3_ can be used as an etching agent for amorphous carbon under electron exposure [57], which would additionally expose further accessible Ru sites that can contribute to C–N bond formation.

To conclude, the described reactions contribute to a complex chemical environment in the deposits during processing that involves multiple sites prone to crosslinking or incorporation of N from NH_3_. The observed C:N ratio of ~3:4 after processing, thus, cannot be assigned to a single chemical mechanism yielding a defined carbon nitride material with known stoichiometry. Rather, the multitude of possible reaction pathways needs to be held responsible for the composition of the obtained material after combined electron and NH_3_ processing. Future studies may aim at a thorough understanding of the dominant chemical features in dependence of the processing conditions. For instance, XPS can yield valuable information on the bonding motifs of C, N, and Ru in the processed deposits. Especially, the role of accessible Ru centers and intermediate species attached to them could be further investigated by this approach.

## 5. Conclusions

Our present study provides evidence that combined NH_3_ and electron processing of deposits fabricated from the FEBID precursor (EtCp)_2_Ru yields deposits with Ru embedded in a material with the chemical composition of a carbon nitride (C_3_N_4_). NH_3_, thus, becomes incorporated in the deposit via the formation of covalent C–N bonds. In the case of pre-purification of the deposits using electron exposure in the presence of H_2_O vapor, the carbon content of the matrix can be reduced before N is incorporated, resulting also in an overall lower C:N ratio. By a combination of H_2_O purification and NH_3_ processing, the stoichiometry of the obtained material can, thus, be tuned. The results of a surface science study on combined NH_3_ and electron processing of (EtCp)_2_Ru model deposits suggest that the electron-induced reactions of NH_3_ with the cyclopentadienyl-type ligands are responsible for the formation of amine species that degrade and crosslink the EtCp ligands under prolonged electron exposure. The process described in this contribution may allow access to the fabrication of a new class of carbon nitride functional nanodevices by FEBID. The affinity of nitrogen as a binding site for e.g., H_2_O is higher than that of pure carbon compounds. Therefore, carbon–nitride devices could in further studies be tested for their ability to outperform existing sensing concepts relying on carbonaceous FEBID deposits.

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
