# Peer review of "Combined Ammonia and Electron Processing of a Carbon-Rich Ruthenium Nanomaterial Fabricated by Electron-Induced Deposition"

_micromachines, 2020, doi:10.3390/mi11080769_

Round 1

Reviewer 1 Report

The authors give a detailed analysis of the electron beam induced reaction of ammonia and water with EBID deposits made from (EtCp)2Ru, both in terms of a nanofabrication as well as a surface science study. In addition to EBIE of carbon from the deposits, they report on the "deposition" of nitrogen into the material to give a composition similar to that of carbon nitrides. A discussion of the possible reaction pathways is given and possible applications as improved sensor materials are briefly discussed.

The manuscript is generally well-written and concise, and the claims are supported by the presented data. Some minor issues are stated below:

Scientific

  • Section 2 / 3.1: The manuscript should clearly state which kind of substrate was used for the FEBID study. Section 2 does not give a material at all, while section 3.1 references an "SiO2 native layer" (L140) and the caption of Figure 1 claims a "Si/SiO2 substrate" (L146). The nature of the substrate is especially relevant as the authors attribute the swelling and bubble formation to the formation of trapped H2 in substrate.
  • Figure 2: The N-map of the H2O-preprocessed material shows little definition and overlap with the C-map, but rather seems to match the O-map. This could indicate a potential decomposition of the NH3 at sites previously etched by H2O, and potentially the formation of a Si-N-O-(C)-containing material. Did the authors test the EBI reaction of NH3 on the pristine (presumably Si w/ native oxide, see above) substrate? Also, the authors should comment on the workflow used to determine the atomic composition of the material as shown on the left (single point scan, average from map, ...).
  • Line 269: the nominal m/z of N2H4 should be 32
  • Section 4: The authors present a detailed mechanistic model (or set of potential reaction pathways) for the formation of CxNy (C:N ~ 3:4) from (EtCp)2Ru, with a focus on the organic chemistry of the carbon- and carbon-nitrogen-based compunds. Less focus is put on the role of the Ruthenium center. As it was reported before (10.1088/0957-4484/23/37/375302) that NH3 can be used to completely remove amorphous carbon via electron beam induced etching, it would be highly interesting to determine the role of the Ru, and whether one should not consider the deposit as RuCxNy rather than "Ru @ C3N4" or just C3N4.

Language, Style

  • L21: "surface [science] study"

Reviewer 2 Report

Rohdenburg and co-authors present their findings on ammonia and electron beam processing of Ru-based FEBID nanostructures, making use of two complementary approaches – ESEM for the direct FEBID approach and UHV for the surface science approach. The authors show that the NH3 and electron beam processing can tailor the composition and morphology of the deposits; in particular, a possibility to form carbon nitrides seems very appealing in a broader view. The authors also outline a plausible reaction mechanism during processing of the FEBID material, which represents a nice wrap-up of what is overall a well-written manuscript. I find these results very interesting and definitely worth publishing.

There are some minor points in the manuscript that could be addressed, just to remove possible ambiguities for the readers:

  1. Lines 96-98: Can you please make it clear what the substrate was in FEBID ESEM experiments? (later in the text one can assume it was some kind of Si/SiO2 chip from the results, but please make it clear here)
    Similarly, in lines 102-104, it would also help to clearly state what the substrate was in UHV experiments (indirectly one can assume it was a Ta sheet, but it is not explicitly clear whether this was just a mount/holder or the actual substrate, just a hint later in line 327).
    If the substrates were not of the same kind in these two approaches, can you please make it clear in the text and provide a brief comment (1-2 sentences) how and if this can affect the comparison of the results and final conclusions?

  1. Page 4: the authors demonstrate the swelling and bubbling effects when the deposits are scanned by an electron beam in presence of NH3 gas. What is missing in my opinion is the effect of scanning on the bare substrate(s), i.e. some kind of control experiments; have the authors checked what happens when the pristine substrate (Si/SiO2, or even Ta) are scanned by an electron beam in the presence of NH3? Is it possible that the swelling/bubbling is (partially) originating from the substrate (in particular this question is initiated by the authors’ claim that the bubbling was mostly present in the halo regions around the main deposit, line 157)

  2. Page 4-6: the EDX characterization. It was not clear when the EDX measurements were taken – during the electron-beam processing or separately, after it? In the latter case, it is known that the electron-beam exposure of the deposits during EDX measurements could to some extent alter the morphology (and possibly composition) of the material (after all, one can think of this as additional electron-beam processing/annealing).
    What were the conditions during these EDX measurements, e.g. the total electron dose, current, acquisition time, even the ESEM chamber pressure…? Can the authors comment if these EDX measurement had any visible effects on the deposits? In particular, the height of the deposits could be altered during the EDX measurements. From the practical point of view, it is simpler to first perform EDX measurements in the ESEM just after deposition and then take the samples out of the ESEM to an AFM; one could assume the authors followed this approach. In that case, can the authors comment whether the EDX measurements could have affected the height of the structures i.e. if AFM results were in some way compromised by the previous EDX characterization?
